# The Quality of Ciders Depends on the Must Supplementation with Mineral Salts

**DOI:** 10.3390/molecules25163640

**Published:** 2020-08-10

**Authors:** Tomasz Tarko, Magdalena Januszek, Aneta Pater, Paweł Sroka, Aleksandra Duda-Chodak

**Affiliations:** Department of Fermentation Technology and Microbiology, Faculty of Food Technology, University of Agriculture in Krakow, ul. Balicka 122, 30-149 Krakow, Poland; magdalena.kostrz@urk.edu.pl (M.J.); Aneta.Pater@urk.edu.pl (A.P.); pawel.sroka@urk.edu.pl (P.S.); aleksandra.duda-chodak@urk.edu.pl (A.D.-C.)

**Keywords:** apple must supplementation, ions, yeast strains, cider

## Abstract

Providing yeast with the right amount of mineral salts before fermentation can contribute to improving the entire technological process, resulting in a better-quality final product. The aim of this study was to assess the impact of apple must supplementation with mineral salts ((NH_4_)_2_SO_4_, MgSO_4_, (NH_4_)_3_PO_4_)) on enological parameters, antioxidant activity, total polyphenol content, and the profile of volatile cider compounds fermented with various yeast strains. Rubin cultivar must was inoculated with wine, cider, and distillery or wild yeast strains. Various mineral salts and their mixtures were introduced into the must in doses from 0.167 g/L to 0.5 g/L. The control sample consisted of ciders with no added mineral salts. The basic enological parameters, antioxidant properties, total polyphenol content, and their profile, as well as the composition of volatile compounds, were assessed in ciders. Must supplementation with magnesium salts significantly influenced the use of the analyzed element by yeast cells and was dependent on the yeast strain. In supplemented samples, a decrease in alcohol concentration and total acidity, as well as an increase in the content of extract and total polyphenols, was observed compared to the controls. The addition of ammonium salts caused a decrease in the amount of higher alcohols and magnesium salts, as well as a decrease in the concentration of some esters in ciders.

## 1. Introduction

Cider is an alcoholic drink made from fresh or concentrated apple juice [1]. For the production of cider, special apple varieties with a characteristic bitter, pungent, and sour taste are most often used [2]. Drinks made from this type of fruit are more aromatic [3]. The cider production process usually takes place as a result of fermentation carried out by selected yeast strains, as well as following spontaneous fermentation, which offers a large variety of microorganisms [3]. During the entire fermentation process, yeast metabolizes the sugar available in apple juice mainly into ethanol and carbon dioxide, in addition to many other compounds that affect the aroma and taste of the drink [1]. The selection of the right yeast cultures necessary for proper fermentation is, therefore, very important when planning the entire alcoholic beverage production process [4]. In addition, yeast affects the polyphenol profile, antioxidant properties, acidity, and ethanol concentrations, which in turn also affect the sensory profile of the final product [5]. It was shown that the use of selected yeast cultures enables the production of a cider which possesses all the desired characteristics [6].

Although the demand for cider is still growing, relatively few scientific studies were conducted characterizing how the addition of mineral compounds affects the quality of these alcoholic beverages. The lack of nutritional balance resulting from a shortage of minerals during must preparation and fermentation prevents the optimal performance of yeast cells during the entire fermentation process [7]. The must’s mineral composition prior to fermentation should include metal ions necessary for yeast cells [8]. In studies carried out by Schmidt [9], the addition of ammonium sulfate to rice bran fermentation with the *Rhizopus oryzae* species caused an increase in biomass and phenolic compounds in the final product. In turn, the studies of Shahirah et al. [10] demonstrated that the addition of magnesium sulfate during fermentation using the yeast strain *Saccharomyces cerevisiae* led to an increase in the bioethanol yield from oil palm trunk juices. Ammonium phosphate added to grape must fermentation contributed to an increase in CO_2_ production efficiency during the whole process [11]. These studies confirm that the addition of individual minerals to the production of alcoholic beverages can contribute to the improvement of the entire technological process, resulting in a better-quality final product.

The aim of the study was to check the effect of apple must supplementation with ammonium sulfate, magnesium sulfate, ammonium phosphate, and mixtures thereof on the use of mineral substances by various yeast strains (winemaking, distilling, cider, and wild yeast strains) during the fermentation of apple must. In addition, the effect of supplementation on enological properties, phenolic compound concentration, antioxidant activity, and the volatile compound profile were also tested.

## 2. Results and Discussion 

### 2.1. The Use of Mineral Substances by Yeast during Apple Must Fermentation

Macro-elements and micro-elements play an important role in the fermentation of alcoholic beverages, contributing to the growth of yeasts and the proper course of their metabolism [12]. The content of metal ions in the must is variable and depends on the type of fruit, their quality, country of origin, weather conditions, agrotechnical treatments, and many more [13]. Before starting the fermentation process, the apple must was analyzed for the content of individual metal ions. The analyzed elements included magnesium, in a concentration of 65.51 mg/L, calcium (26.58 mg/L), and zinc (0.71 mg/L). Walker [14] and Birch and Walker [15] proved that magnesium is a factor protecting against yeast stress in the production of alcoholic beverages and can help prevent cell death caused by temperature shock and ethanol toxicity. Magnesium ions are required as a cofactor by over 300 enzymes, including those necessary for glycolytic, alcohol, and fatty acid biosynthesis [16]. The results obtained were similar to those of Eisele and Drake [17] for apple juice. In order to check the effect of metal ions on the course of the fermentation process in the presence of various yeast strains, apple must was further enriched with minerals (ammonium sulfate, magnesium sulfate, ammonium phosphate, and mixtures thereof). Magnesium content in all fermented samples was in the range of 45.5 mg/L–165.5 mg/L (Table 1, Table 2, Table 3 and Table 4). Must supplementation with magnesium sulfate significantly increased the utilization of the analyzed element (31.3%) by wild yeast, compared to pre-fermentation samples (Table 4). Ciders supplemented with magnesium sulfate, then fermented with distillery or wine yeast, contained similar amounts of magnesium ions, at the level of 165 mg/L (Table 1 and Table 2). In tests where magnesium sulfate and ammonium sulfate were added, a greater use of magnesium ions (a 29.18% increase) by cider yeast was observed, compared to the control (Table 3). The use of minerals depends largely on the yeast strain used. Increasing magnesium content may provide some level of protection for yeast exposed to osmotic stress by maintaining the structural integrity of membranes. In turn, excess magnesium in cells contributes to some mechanisms of repair or improvement of cellular functions [15]. Another element analyzed was calcium, which participates in the enzymatic reactions of yeast cells [18]. Calcium binds to yeast cell walls and plays a key role in flocculation, which is important during the fermentation process [19]. Calcium in all analyzed ciders was in the range 21.73 mg/L–52.3 mg/L (Table 1, Table 2 and Table 3). In musts supplemented with ammonium phosphate or ammonium sulfate with magnesium sulfate, all yeast strains used about 19% more calcium, compared with samples without supplementation. In ciders with the addition of ammonium sulfate, a significant increase in the analyzed element was observed after the end of the fermentation process. In samples fermented with distillery yeast, there was up to 95.3% more calcium than in samples before fermentation (Table 2). The last element analyzed was zinc. Zinc ions are an essential nutrient for yeast, needed for the proper course of the fermentation process. The optimal demand for this element by yeast during fermentation is in the range of 0.1 mg/L^–1^ mg/L [16]. The observed average use of zinc by four tested yeast strains was in the range of 12%–28% compared to the samples before fermentation (Table 1, Table 2, Table 3 and Table 4). Furthermore, in this case, the addition of ammonium sulfate alone caused a significant increase in the zinc content in all samples after the fermentation process was completed. The largest increase was observed in samples fermented with cider yeast, even by 100%, compared to samples before fermentation (Table 3).

The observed increase in the amount of calcium and zinc in supplemented samples may be associated with the presence of pectins in ciders. Apples are very rich in this polysaccharide. These compounds have a complex structure, mainly consisting of linear homogalacturonate chains, made of galacturonic acid residues connected by an α-(1,4) bond, some of which can be esterified with methanol. Pectins are polyelectrolytes because they contain ionizing functional groups in the molecule. Free carboxyl groups can dissociate and interact with monovalent and polyvalent ions present in a solution and react with cations of other polyelectrolytes, e.g. amino acids, peptides, and proteins [20]. Pectinomethylesterase found in commercial pectinolytic preparations catalyzes the hydrolysis of pectin, esterifying methylene groups, as well as increasing the number of free carboxyl groups and the ion exchange capacity of pectin molecules. The enzyme action releases carboxyl groups that can interact with cations found in the must [21]. Protons and monovalent ions contained in the anionic polyelectrolyte structure can be exchanged for multivalent cations such as Ca^2+^, Mg^2+^, and Zn^2+^, causing crosslinking bridges connecting different fragments of anionic polyelectrolyte molecules (R-COOH) [22].
R-COOZn^+^ + Mg^2+^ ⇆ R_1_-COO-Mg-OOC-R_2_ + 2H^+^(1)

The increase in the concentration of zinc and calcium ions in settings supplemented with magnesium ions can be explained by the ion exchange process. Zinc and calcium ions occurring naturally in apple must are bound by polyelectrolyte molecules present in the fermenting medium. The addition of magnesium ions to the must shifts the thermodynamic equilibrium and, according to the equilibrium law, an exchange reaction of multivalent ions occurs (example shown for zinc ions).
R-COOZn^+^ + Mg^2+^ ⇆ R-COOMg^+^ + Zn^2+^(2)

Pectins and other polyelectrolytes found in fermenting must, such as peptides and proteins, also react with each other and with phenolic compounds [20,23], forming insoluble molecules co-precipitating with yeast cells during flocculation. This mechanism reduces the concentration of zinc ions in post-fermentation samples not supplemented with other ions. An adequate nutrient content is important to maximize yeast metabolic activity. Too low a content of magnesium, calcium, or zinc causes stress in yeast cells [24]. Therefore, before starting the fermentation process, it is extremely important to ensure the right amount of minerals necessary for the proper operation of yeast. The proper content of ingredients in the must contributes to yeast conducting a more effective fermentation process, thus contributing to a final product which possesses the desired characteristics [25]. Our research proves that the addition of either magnesium sulfate or ammonium sulfate significantly improves the fermentation process by individual yeast strains. 

### 2.2. Influence of Minerals on the Enological Properties of Ciders

An important quality parameter for cider is ethanol concentration. This is a key factor affecting the quality, sensory properties, and stability of beverages [26]. The alcohol content in ciders obtained by alcoholic fermentation of apple must should be 1.2–8.5% vol. [27]. The ethanol concentration in all analyzed ciders was in the range of 5.7–7.7% vol. (Table 5, Table 6, Table 7 and Table 8). Almost all supplemented trials showed a decrease in alcohol concentration compared to the control. The highest reduction in ethanol content was found in ciders with the addition of ammonium sulfate and ammonium phosphate, and the smallest in ciders supplemented with a mixture of ammonium sulfate, magnesium sulfate, and ammonium phosphate. The type of yeast strain used for fermentation was not statistically significant in this case. Yeast exposed to osmotic stress, caused primarily by an increased concentration of sugars, but also metal ions (ionic stress), synthesizes higher amounts of glycerol [28]. Under extremely unfavorable conditions, the amount of glycerol produced can be up to 10,000 times higher than in samples with optimal composition [29]. It was shown [30] that yeast synthesis of amino acids from free amine nitrogen influences the production of higher glycerol concentrations. Glycerol metabolism is crucial in adapting yeast to changed environmental conditions [31]. The use of sugars to produce glycerol results in lower ethanol levels in these tests (Table 5, Table 6, Table 7 and Table 8). The addition of magnesium ions increases the tolerance of yeast cells to ethanol during fermentation [12,32]. However, the ciders examined in this study are probably too low in alcohol content to observe a similar effect.

The content of general extract is directly related to the content of ethyl alcohol and residual sugars. Usually, a higher concentration of ethanol suggests less sugars remaining after fermentation and a smaller total extract [33]. The value of this parameter in the analyzed ciders was in the range of 16 g/L–22.5 g/L (Table 5, Table 6, Table 7 and Table 8). It was shown that the total extract content was higher in all supplemented ciders than in the control samples. These values correspond to the alcohol content, whereby a higher concentration of the extract denotes lower cider strength. The smallest effect of supplementation was observed in samples fermented with wild and cider yeast (Table 7 and Table 8). The higher content of the total extract than in the control samples was probably related to the glycerol synthesis described above. Glycerol is an important component of ciders and wines, affecting their sugar-free extract and determining the full flavor of the drink.

The total acidity of ciders should be between 3.5 g and 7 g of malic acid per liter [34]. The initial acidity of apple juice before fermentation was 2.01 g/L. In the control samples after the end of the fermentation process, the total acidity decreased, ranging from 1.2 g/L–1.7 g/L of malic acid. The addition of ammonium sulfate caused the smallest decrease in the total acidity compared to the initial acidity of apple must (Table 5, Table 6, Table 7 and Table 8). All the analyzed samples had an overall acidity much lower than recommended by Polish law regulating ciders. The low acidity of ciders is associated with the use of the Rubin apple variety in the production process. These apples are most often used for direct consumption and their acidity is low. The volatile acidity of ciders should be less than 0.9 g/L, calculated as acetic acid [34]. All ciders analyzed met this requirement. In ciders obtained with cider yeast or wild yeast, the addition of minerals did not significantly affect volatile acidity (Table 5, Table 6, Table 7 and Table 8). In samples fermented by wine and distiller’s yeast, the addition of either magnesium sulfate or ammonium phosphate slightly increased the value of volatile acidity compared to other samples.

An essential nutrient for yeast growth and metabolism during fruit juice fermentation is the appropriate content of free nitrogen compounds (FAN) [35,36]. The relative content of nitrogen compounds varies depending on the species and variety of fruit. Limited information is available on the concentration of FAN in apple must and how this concentration affects cider fermentation. In studies conducted by Alberti et al. [37] FAN concentration in 51 apple samples from Brazil averaged 38.3 mg/L. In turn, other studies conducted by Valois et al. [38] showed that the FAN content in apple juice obtained from apples from New York was in the range of 12 mg/L–190 mg/L, and that in Polish apples of Rubin variety was 72.45 mg/L [6]. In our studies, the content of free amine nitrogen in the analyzed must was low and amounted to 51.58 mg/L. The content of nitrogen compounds is, therefore, dependent on the quality parameters of fruits such as apple varieties and environmental conditions, including exposure to the sun or fertilization method and harvest time [39]. After fermentation, the FAN content in the obtained ciders was 40 mg/L–60 mg/L (Table 5, Table 6, Table 7 and Table 8). Metabolically active distillery yeast without must supplement used the highest nitrogen content (40%) compared to the initial sample (Table 6). Must supplementation with ammonium phosphate increased the FAN content in the analyzed samples. The largest increase (by 23%) was observed for samples fermented with cider yeast (Table 7). Must supplementation with ammonium sulfate also contributed to an increase in the content of free nitrogen compounds. The highest FAN content (increase by 26.4% compared to the control) in this case was observed in samples fermented with wine yeast (Table 5). Supplementation of must with mineral ingredients significantly contributed to the increase in the content of free nitrogen compounds (FAN) in the obtained ciders. In studies conducted by Slininger et al. [32], supplementation caused a significant interaction between nitrogen compounds and mineral compounds, which was key to maintaining proper fermentation.

Polyphenols are an important group of bioactive compounds because they have a strong impact on the quality of food products. These compounds are also powerful antioxidants with a wide range of biological activities [40,41]. Particular attention is paid to the content and profile of polyphenols when choosing the apple variety used for cider production, as they affect the color and astringency, which make up sensory perceptions [42]. In studies conducted by Riekstina-Dolge et al. [43], the total polyphenol content in dessert apples was on average 27.78 mg/L–92.32 mg/L. A higher content of these compounds was observed in ciders made from cider apples (69.9 mg/L–92.32 mg/L). In the analyzed ciders produced with various yeast strains with the addition of minerals, the content of polyphenols ranged from 11.8 mg catechin/100 mL–18.9 mg catechin/100 mL (Table 5, Table 6, Table 7 and Table 8), which is characteristic of the apple variety used [6]. The addition of minerals in most samples significantly increased the content of polyphenols. The largest increase was observed in ciders fermented with wine, distillery, or cider yeast, where the addition was ammonium sulfate, magnesium sulfate, and ammonium phosphate as a salt mixture (16.7 mg catechin/100 mL–18.9 mg catechin/100 mL) compared to control samples (12.3 mg catechin/100 mL–13.4 mg catechin/100 mL). The lowest content of polyphenols was observed in all samples fermented with wild yeast; in this case, the addition of minerals slightly increased the content of these compounds, but these differences were not statistically significant (11.8 mg catechin/100 mL–13.1 mg catechin/100 mL). The obtained polyphenol content results depend to a large extent on the yeast strains used, on additions of mineral elements in this case, and on the time of apple harvesting, as well as on the technology and conditions for the preparation of ciders [6]. In a study carried out by Schmidt and Furlong [9], a similar relationship was observed in the increase of phenolic compounds as a result of the addition of ammonium sulfate to the must (during and before fermentation).

Minerals are important cider components due to the share of these elements in oxidation processes, as well as changes in color and stability of the final product [44]. The content of polyphenols also directly affects the antioxidant properties of various products [45]. Knowledge of antioxidant activity can help improve processing technology and fruit storage conditions, as well as help consumers make better choices in terms of product quality [46]. Cider fermented with cider yeast was characterized by the highest antioxidant activity (Table 7). A significant increase, compared to the control, was also observed with the addition of a mixture of ammonium sulfate and ammonium phosphate (96.4 mg of Trolox/100 mL) (Table 5). Ciders obtained during fermentation using wild yeast had the smallest increase in antioxidant activity compared to other samples (Table 8). The obtained results showed that ciders made from Polish apple varieties contain a lower amount of polyphenols compared to ciders from other European countries; however, due to their antioxidant properties, which were significantly increased by the addition of minerals, they may have a beneficial effect on consumer health.

### 2.3. The Effect of Mineral Addition on Volatile Compounds in Ciders

One of the key purposes of using different yeast strains and apple must supplementation in cider production is to modify the sensory profile. The aromas of alcoholic beverages are mainly influenced by esters, higher alcohols, terpenes, and acids [47]. The analysis of volatile compounds provides information on the raw materials used and the entire technological process [48]. The most important precursors and intermediates for biosynthesis of many volatile compounds are amino acids found in musts [49]. Higher alcohols are the main group of volatile compounds in fermented beverages. They arise during fermentation from catabolism of keto acids, including the degradation of amino acids (valine, leucine, isoleucine, threonine, and phenylalanine) or through catabolism of amino acids via the Ehrlich pathway [50]. After alcoholic fermentation in the tested samples, isobutanol, 3-methyl-butanol, and 2-methyl-butanol were determined in the alcohol group. The formation of certain higher alcohols (giving a pleasant fruity smell) from specific amino acids was proven; for example, 3-methyl-1-butanol is formed from leucine, 2-methyl-1-butanol from isoleucine, and isobutanol from valine [51]. In the analyzed ciders, it was observed that the addition of some minerals contributed to the reduction of higher alcohol content, compared to the control samples, especially in the case when musts were supplemented with the addition of ammonium salts (Table 9, Table 10, Table 11 and Table 12). The exception was isobutanol, whose concentration was significantly higher in samples supplemented with magnesium sulfate and ammonium phosphate (Table 9, Table 10, Table 12). Kotarska and Dziemianowicz [52] found that the concentration of the total higher alcohol content was halved in spirits obtained from fermentation of molasses worts stimulated by the addition of ammonium phosphate (V) or with mixtures, i.e., ammonium sulfate (VI), ammonium phosphate (V), and (VI) magnesium sulfate or ammonium phosphate (V) and calcium pantothenate. Based on the research carried out by Arrizon and Gschaedler [53], as a result of adding ammonium sulfate and amino acids during the tequila fermentation process, a higher concentration of propanol and a lower concentration of isobutanol, amyl alcohols, and methanol compared to the control samples were found. Higher alcohols can be produced by catabolic conversion of the branched-chain amino acids (Ehrlich pathway) or by the anabolic formation of these amino acids de novo from sugars. The increase in production of higher alcohols is a result of low assimilable nitrogen, as more carboxylic acids (α-cetoacids) are available for the synthesis of higher alcohols than for amino acid production [53,54]. In our case, with supplementation, it is possible that the synthesis of higher alcohols occurred via anabolic formation of amino acids. Since the yeast cell requires the presence of amino acids to form higher alcohols and the corresponding esters responsible for the fruit aroma, it is important to ensure that yeast cells have access to the right amount of nitrogen compounds [55,56,57]. Despite the observed decrease in the concentration of higher alcohols, the highest content of isobutanol, compared to other samples, was observed in ciders fermented with distillery yeast Ethanol RED (*S. cerevisiae)* (Table 9). The obtained results are confirmed in the literature, which states that the yeast *S. cerevisiae* synthesizes large amounts of isobutanol in comparison with other yeast strains [58]. Research conducted by Arslan et al. [59] confirms that the distillery yeast strain used significantly affects the quality of the product obtained and the composition of the aroma during fermentation. These effects include an increase in the concentration of some higher alcohols, including isobutanol. In other ciders the obtained isobutanol values were similar to those obtained by Loira et al. [60]. This shows that not only the yeast strain used has a significant impact on the concentration of volatile compounds, but also that the addition of minerals contributed to obtaining a similar value of higher alcohols in the analyzed ciders compared to the results obtained by other researchers [61]. Magnesium ions play a fundamental role in the growth and metabolism of yeast cells. Magnesium ions are required as a cofactor by over 300 enzymes, including those essential for the activity of glycolytic, alcohologenic, incorrect ref order, 67 detected after 38. You jumped the numbers in betweenand fatty acid biosynthesis. They stabilize biological membranes and are important for nucleic acids, ribosomes, polysaccharides, and lipids (stress suppressors). The resultant effects of stress on yeast cells during fermentation causes disproportionate synthesis of esters and higher alcohols [62,63].

Esters are the main group of desirable compounds that shape the aroma of alcoholic beverages. Esters are produced by yeast during the fermentation process in the reaction between alcohols and fatty acids with coenzyme A (CoASH) and ester synthesizing enzymes [64]. The most important esters in fermented beverages are ethyl 3-methylbutyrate, 2-phenylethyl acetate, ethyl hexanoate, ethyl octanoate, and ethyl decanoate [65]. Ethyl acetate is particularly important for achieving the desired sensory perceptions in the finished product. In low concentration, ethyl acetate is associated with fruity, sweet, grape, and rum aromas, while, in a concentration above 150 mg/L–200 mg/L, it can cause an unpleasant cider smell, associated with the smell of nail polish [66,67]. Ethyl acetate was the dominant ester in the analyzed ciders (Table 9, Table 10, Table 11 and Table 12), but its value was below 150 mg/L in all trials. Supplementation with mineral components significantly contributed to the reduction of ethyl acetate content in fermented ciders with distillery, wine, and cider yeast, compared to the control samples (Table 9, Table 10, Table 12). Different results were presented by Arizon and Gschaedler [53], showing a higher concentration of ethyl acetate due to the addition of ammonium sulfate and amino acids during the tequila fermentation process. Fujiwara et al. [68] claimed that ester synthesis increases as available amino nitrogen increases. Beltran et al. [69] observed that nitrogen addition in the early stage of fermentation using a mixture of ammonium and amino acids in synthetic grape must resulted in an increase in ester synthesis. The decrease in the concentration of ethyl acetate in the samples we analyzed could be associated with a decrease in the concentration of higher alcohols, because the main enological importance of higher alcohols is that they are precursors of acetate esters [70]. Over-production of ethyl acetate could be associated with excessive addition of YAN (yeast assimilable nitrogen), which is the sum of the nitrogen content of primary amino acids and ammonium in musts [71]. However, a higher content of ethyl acetate was observed in samples fermented with wild yeast (Table 11). Satora and Tuszyński [72] found that wild yeast, such as *Candida pulcherrima,* which start spontaneous fermentation of apple wines, produce very large amounts of ethyl acetate (200 mg/L), while *Saccharomyces* strains, produce only a fraction of this compound (2 mg/L). Similar results were presented by Patelski et al. [73], showing a significantly higher concentration of ethyl acetate in plum distillates obtained from spontaneously fermented musts (299.76 mg/L 100°) than with the yeast *S. bayanus* (123.66 mg/L 100°). The results obtained were similar to those of Madrera et al. [74]. In most cases, as a result of supplementation, the concentration of the remaining esters (isobutyl acetate, isopentyl acetate, ethyl hexanoate, and hexyl acetate) increased in comparison with the control samples. These compounds bring pleasant fruity aromas to the liquors, e.g., isobutyl acetate has a fruity (currant/pear), floral (hyacinth/rose) aroma, whereas isoamyl acetate has a fruity, banana, sweet, fragrant, powerful aroma with a bittersweet taste reminiscent of pear. Ethyl hexanoate has a powerful, fruity aroma with a pineapple/banana note, while hexyl acetate has a pleasant fruity, apple, cherry, pear, floral aroma [75] (Table 9). The nitrogen composition of the must may have a strong impact on the accumulation of volatile esters during the fermentation process. Esters tend to increase in line with must amino nitrogen, although the responses of the various esters to individual amino acids is unknown. The addition of ammonium salts strongly stimulates the production of esters [76]. Another interesting phenomenon was the absence of ethyl lactate in controls and its presence in supplemented samples. Furthermore, the highest concentration of ethyl lactate was found in ciders supplemented with ammonium sulfate (even several times higher than the concentration compared to supplementation with other compounds). This may be due to the fact that the addition of ammonium sulfate promotes the production of lactic acid, which in turn is formed by the esterification reaction of ethanol and lactic acid during fermentation [77,78]. Ethyl lactate has a light ethereal, buttery aroma [75].

## 3. Materials and Methods 

### 3.1. Materials

#### Ciders Preparation

Apples (Rubin cultivar) used in the experiments were obtained from a pomological orchard of the University of Agriculture, located in Garlica Murowana (near Cracow, Poland). The apples were washed, ground, and pressed on a Zottel hydraulic press (35 L). The musts (0.5 L) obtained from particular cultivars were poured into bottles (0.7 L) and supplemented with ammonium sulfate (SA), heptahydrate magnesium sulfate (values based on the anhydrous salt) (SM), and ammonium phosphate (FA) in the amount of 0.5 g/L, as well as mixtures of SA + SM, FA + SA, and SM + FA salts (0.25 g/L + 0.25 g/L, respectively) and SA + SM + FA salts (0.167 g/L + 0.167 g/L + 0.167 g/L). Must was inoculated with various yeast strains (Erbslöh, Geisenheim, Germany): wine (Elegance), cider (Gozdawa), distillery (Ethanol RED), or wild (Wild & Pure) at 0.2 g/L (dry yeasts were hydrated according to the manufacturer’s recommendations). The bottles with inoculated must were closed with glycerine-filled fermentation tubes, and then allowed to ferment for two weeks at 20 °C. The fermented musts were drained from the yeast sludge and aged at 4 °C for three weeks.

### 3.2. Methods

#### 3.2.1. Determination of Total Acidity, Volatile Acidity, Total Extract Content, and Ethyl Alcohol Content

The determination of total extract content, ethyl alcohol content, total acidity, and volatile acidity was conducted in accordance with the methods recommended by the International Organization of Vine and Wine [79]. Total extract content and ethanol content in ciders were determined by distillation methods using pycnometric density determination, total acidity was determined with the potentiometric method, and volatile acidity was determined using the titration method.

#### 3.2.2. Free Amino Nitrogen (FAN) Content 

Free amino nitrogen (FAN) was determined with the ninhydrin method. The absorbance of the samples was measured at a wavelength λ = 575 nm [80].

#### 3.2.3. Antioxidant Activity

The antioxidant capacity of samples was determined by the ABTS-cation radical scavenge assay (diammonium salt of the 2,2′-azino-*bis*-(3-ethylbenzothiazoline-6-sulfonic) acid) [80]. Active ABTS was produced through a chemical reaction with potassium persulfate. Absorbance was measured spectrophotometrically at 734 nm. The results obtained were compared with the ability to scavenge ABTS radical by Trolox (synthetic vitamin E) and expressed as mg Trolox/100 mL (TEAC, Trolox equivalent antioxidant capacity). Absorbance measurements were performed on an ultraviolet–visible light (UV–Vis) spectrophotometer Beckman (type DU 650, Warsaw, Poland). 

#### 3.2.4. Total Polyphenol Content 

Total polyphenol content was determined using a spectrophotometric method (UV–Vis Beckman) with Folin–Ciocalteu reagent [39]. The results of the total polyphenols were expressed as mg catechin/100 mL, based on a standard curve.

#### 3.2.5. Analysis of Volatile Compounds 

A sample of 2 mL was placed into a 15-mL headspace vial, and 50 μL of internal standard solution (5 mg/L of ethyl nonanoate) was added. Then, the SPME (solid-phase microextraction) fiber (85 μm Carboxen Polydime thylsiloxane, Supelco, St. Louis, MO, USA) was placed in the headspace above the sample, and the vial was incubated for 30 min at 40 °C. The fiber was subjected to thermal desorption in a gas chromatograph injector at 250 °C.

The chromatographic separation was carried out on a Clarus 580 apparatus (PerkinElmer, Waltham, MA, USA) and a Crossbond dimethyl polysiloxane 60 m, 0.25 mm, 1.4 μm film thickness column (Restek, MA, Pennsylvania, USA). The carrier gas flow (He) was 2 mL/min, with the following temperature program: 35 °C, 6 min; 8 °C/min up to 180 °C; 12 °C/min up to 220 °C; 25 min. The detector and dispenser temperature was 250 °C. An HT2800T autosampler (HTA Brescia, Italy) was used, and PerkinElmer Total Chrom 6.3.2 software (PerkinElmer, Waltham, MA, USA) was used to integrate the results.

#### 3.2.6. Absorptive Atomic Spectrometry (ASA)

##### Mineralization

Before analyzing metal ions, must and cider samples were mineralized to achieve the total decomposition of their organic substances. Samples of 2 mL were added to mineralizing dishes, and concentrated nitrogen acid (V) was poured over them (65%, 3 cm²). The samples were wet mineralized in a Mars Express microwave oven (at a maximum temperature of 170 °C, duration: 40 min). When wet mineralization was completed, the content of the mineralization dishes was transferred quantitively to test tubes and supplemented with deionized water to reach 14 cm³.

##### Determination of Metal Ions

Magnesium, calcium, and zinc were analyzed with a VARIAN 240FS spectrometer using the atom spectrometer absorption method with flame atomization (air/acetylene). The device uses an automatic sample dosing system SPIS-20. Mg^2+^ ion absorbency was determined at a wavelength of 202.6 nm, while wavelengths of 422.7 nm and 213.9 nm were used for Ca^2+^ and Zn^2+^, respectively. Fast sequential mode was run for the determination during a single aspiration of a sample.

#### 3.2.7. Statistical Analysis 

The results presented in the study were the means of three independent repetitions with determination of the standard deviation. The data were analyzed with variance analysis (ANOVA) in order to establish the significance of the tested parameters. Statistically significant differences between the means were verified with a Tukey’s set using Statistica 10 statistical software (StatSoft Polska, Cracow). 

## 4. Conclusions

Studies show that supplementing apple must with mineral salts ((NH₄)₂SO₄, MgSO_4_, (NH_4_)_3_PO_4_)) and the use of different yeast strains can affect the quality parameters and profile of cider aromas. It was observed that the addition of selected minerals can cause a decrease in alcohol concentration and acidity, but an increase in the amount of total extract in ciders. The lowest concentration of ethanol, compared to the control samples, was observed in ciders with the addition of either ammonium sulfate or ammonium phosphate. The increase in extract content was inversely proportional to the alcohol concentration in ciders. Supplementation of must with mineral ingredients significantly contributed to the increase in the antioxidant activity of these drinks; this effect was also dependent on the yeast strains used for fermentation. In addition, it was shown that the content of volatile compounds (higher alcohols and esters) in ciders is influenced not only by the type of fermentation used and the conditions in which it is performed, but also by earlier must supplementation with minerals. Ammonium salts contribute to lowering the amount of higher alcohols in the finished product, while magnesium salts cause lower concentrations of some esters. To sum up, it can be said that the addition of appropriate ions during the production of ciders can affect both the physico-chemical characteristics (alcohol content, FAN, antioxidant activity) and the profile of volatile compounds of alcoholic beverages. Summing up, taking into account the enological parameters of the obtained ciders and the profile of volatile compounds, it can be concluded that the most desirable qualitative effects are achieved by supplementation of must with ammonium sulfate and ammonium phosphate (0.25 g/L + 0.25 g/L) and by conducting the fermentation with wine yeast.

## Figures and Tables

**Table 1 molecules-25-03640-t001:** The content of selected ions in ciders fermented by wine yeast (W), with the following additions: SA—ammonium sulfate, SM—magnesium sulfate, FA—ammonium phosphate.

Sample/Ions	Ca mg/L	Zn mg/L	Mg mg/L
Control	29.3 a (±1.03)	0.65 a (±0.06)	67.00 a (±0.69)
SA	46.6 b (±2.57)	1.32 b (±0.02)	90.62 b (±2.91)
SM	37.04 c (±2.07)	1.01 b (±0.07)	164.97 c (±1.54)
FA	21.73 d (±1.53)	0.73 c (±0.07)	46.00 a (±1.00)
SA + SM	21.73 d (±1.53)	0.82 c (±0.22)	90.44 b (±18.14)
SA + FA	24.62 ad (±4.74)	0.58 c (±0.03)	53.69 a (±4.59)
SM + FA	28.13 ad (±3.43)	0.77 c (±0.04)	102.94 bd (±10.02)
FA + SA + SM	28.36 ad (±0.65)	1.07 c (±0.27)	93.39 b (±1.65)
^1^ Sig.	***	***	***

^1^ Sig.: significance; *** display the significance at 0.5% by least significant difference. Values with different roman letters (a–d) in the same row are significantly different according to the Tukey range test (*p* < 0.05).

**Table 2 molecules-25-03640-t002:** The content of selected ions in ciders fermented by distiller’s yeast (G), with the following additions: SA—ammonium sulfate, SM—magnesium sulfate, FA—ammonium phosphate.

Sample/Ions	Ca mg/L	Zn mg/L	Mg mg/L
Control	34.9 a (±7.47)	0.71 a (±0.19)	71.2 a (±12.54)
SA	51.9 b (±3.47)	1.27 b (±0.02)	89.4 a (±0.43)
SM	52.53 b (±3.17)	0.86 ab (±0.05)	165.49 b (±2.54)
FA	22.8 c (±1.01)	0.84 ab (±0.07)	48.07 c (±0.32)
SA + SM	22.94 c (±0.32)	0.74 a (±0.18)	81.03 a (±1.45)
SA + FA	21.66 c (±1.1)	0.51 a (±0.08)	62.33 a (±4.34)
SM + FA	32.16 ac (±3.21)	0.58 a (±0.01)	103.4 d (±10.99)
FA + SA + SM	27.54 ac (±4.23)	1.06 ab (±0.41)	84.32 ad (±9.31)
^1^ Sig.	***	**	***

^1^ Sig.: significance; ** and *** display the significance at 1% and 0.5% by least significant difference. Values with different roman letters (a–d) in the same row are significantly different according to the Tukey range test (*p* < 0.05).

**Table 3 molecules-25-03640-t003:** The content of selected ions in ciders fermented by cider yeast (C), with the following additions: SA—ammonium sulfate, SM—magnesium sulfate, FA—ammonium phosphate.

Sample/Ions	Ca mg/L	Zn mg/L	Mg mg/L
Control	32.3 a (±2.23)	0.7 a (±0.05)	68.6 a (±3.31)
SA	49.54 b (±3.51)	1.48 b (±0.1)	90.6 b (±2.56)
SM	35.44 a (±3.67)	1.02 ab (±0.1)	161.2 c (±4.44)
FA	22.03 c (±1.93)	0.79 a (±1.18)	45.6 d (±0.76)
SA + SM	21.08 c (±0.44)	0.63 a (±0.18)	76.3 a (±4.32)
SA + FA	24.77 c (±1.2)	0.61 a (±0.06)	52.8 c (±5.19)
SM + FA	31.1 a (±2.76)	0.66 a (±0.13)	110.5 d (±7.62)
FA + SA + SM	30.65 a (±0.07)	0.92 a (±0.33)	95.6 b (±1.48)
^1^ Sig.	*******	*******	*******

^1^ Sig.: significance; *** displays the significance at 0.5% by least significant difference. Values with different roman letters (a–d) in the same row are significantly different according to the Tukey range test (*p* < 0.05).

**Table 4 molecules-25-03640-t004:** The content of selected ions in ciders fermented by wild yeast (D), with the following additions: SA—ammonium sulfate, SM—magnesium sulfate, FA—ammonium phosphate.

Sample/Ions	Ca mg/L	Zn mg/L	Mg mg/L
Control	30.55 a (±1.32)	0.87 ab (±0.02)	65.7 ac (±0.34)
SA	46.7 b (±1.57)	1.31 ab (±0.25)	89.4 a (±2.03)
SM	24.35 a (±0.32)	1.17 ab (±0.06)	113.69 b (±1.82)
FA	22.1 c (±1.23)	0.86 ab (±0.09)	47.94 ac (±0.83)
SA + SM	28.1 ac (±4.3) a	0.65 a (±0.21)	96.96 a (±33.01)
SA + FA	26.22 ac (±3.13)	0.8 a (±0.15)	56.78 ac (±6.05)
SM + FA	30.79 a (±3.24)	0.84 a (±0.23)	105.36 b (±10.59)
FA + SA + SM	28.78 ac (±1.22)	0.89 ab (±0.12)	94.25 a (±3.21)
^1^ Sig.	*******	******	*******

^1^ Sig.: significance; **, and *** display the significance at 1% and 0.5% by least significant difference. Values with different roman letters (a–c) in the same row are significantly different according to the Tukey range test (*p* < 0.05).

**Table 5 molecules-25-03640-t005:** Physicochemical parameters for ciders fermented by wine yeast (W), with the following additions: SA—ammonium sulfate, SM—magnesium sulfate, FA—ammonium phosphate.

	Free Amino Nitrogen (FAN) (mg/L)	Total Polyphenol Content (mg of Catechin/100 mL)	Antioxidant Activity (mg of Trolox/ 100 mL)	Total Acidity (g of Malic Acid/L)	Volatile Acidity (g Acetic Acid/L)	Ethyl Alcohol Content (%)	Total Extract Content (g/L)
Control	39.9 a (±5.56)	13.4 a (±0.31)	66.5 a (±2.79)	1.25 a (±0.22)	0.28 a (±0.04)	6.86 a (±0.04)	16.56 a (±0.75)
SA	57.6 b (±0.78)	17.02 b (±0.64)	87.1 c (±1.53)	2.31 b (±0.06)	0.13 a (±0.01)	5.76 b (±0.07)	21.4 b (±0.00)
SM	47.2 a (±2.12)	14.5 a (±0.25)	63.9 a (±2.31)	1.49 a (±0.07)	0.33 b (±0.03)	7.72 b (±0.11)	18.5 c (±0.25)
FA	60.6 b (±2.23)	15.4 a (±0.5)	74.6 b (±1.92)	1.75 b (±0.05)	0.4 b (±0.09)	5.71 b (±0.04)	18.7 c (±0.17)
SA + SM	48.16 c (±0.86)	15.15 a (±0.5)	74.7 b (±5.4)	1.45 a (±0.07)	0.3 a (±0.03)	6.02 c (±0.12)	18.5 c (±0.8)
SA + FA	65.2 b (±1.49)	17.1 b (±0.9)	96.4 e (±2.42)	2.19 b (±0.05)	0.13 a (±0.02)	6.02 c (±0.04)	22.5 b (±0.58)
SM + FA	46.9 ac (±4.03)	15.5 a (±0.38)	90.2 c (±0.15)	1.36 a (±0.25)	0.26 a (±0.02)	6.3 d (±0.08)	19.4 c (±0.17)
FA + SA + SM	48.3 c (±2.13)	18.9 b (±1.8)	86.4 c (±1.8)	1.82 b (±0.17)	0.15 a (±0.13)	6.17 cd (±0.04)	21.4 b (±0.29)
^1^ Sig.	*******	*******	*******	*******	******	*******	*******

^1^ Sig.: significance; ** and *** display the significance at 1% and 0.5% by least significant difference. Values with different roman letters (a–d) in the same row are significantly different according to the Tukey range test (*p* < 0.05).

**Table 6 molecules-25-03640-t006:** Physicochemical parameters for ciders fermented by distiller’s yeast (G), with the following additions: SA—ammonium sulfate, SM—magnesium sulfate, FA—ammonium phosphate.

	Free Amino Nitrogen (FAN) (mg/L)	Total Polyphenol Content (mg of Catechin/100 mL)	Antioxidant Activity (mg of Trolox/ 100 mL)	Total Acidity (g of Malic Acid/L)	Volatile Acidity (g Acetic Acid/L)	Ethyl Alcohol Content (%)	Total Extract Content (g/L)
Control	30.8 ac (±4.9)	12.3 a (±0.5)	57.2 a (±5.67)	1.74 a (±0.52)	0.28 a (±0.03)	6.84 a (±0.00)	16.46 a (±1.32)
SA	51.5 bc (± 3.3)	16.4 b (±0.2)	82.7 b (±1.9)	2.41 b (±0.05)	0.21 a (±0.00)	5.79 b (±0.04)	21.36 b (±0.25)
SM	32.6 ac (±1.06)	13.6 a (±0.5)	68.4 c (±1.1)	2.19 a (±0.05)	0.26 a (±0.02)	5.84 b (±0.00)	20.86 b (±0.25)
FA	58.2 b (±3.32)	15.7 b (±0.2)	80.4 b (±0.8)	1.74 a (±0.1)	0.36 b (±0.06)	5.84 b (±0.00)	18.23 c (±0.46)
SA + SM	36.8 ac (±7.9)	14.4 c (±0.5)	74.8 bc (±1.6)	2.19 a (±0.08)	0.21 a (±0.03)	5.97 c (±0.16)	20.66 b (±0.75)
SA + FA	54.3 b (±0.52)	16.6 b (±0.5)	94.03 d (±3.3)	2.4 b (±0.05)	0.23 a (±0.04)	5.92 b (±0.00)	22.16 b (±0.25)
SM + FA	39.5 ac (±1.7)	15.4 b (±0.4)	82.7 d (±3.3)	2.36 b (±0.04)	0.27 a (±0.00)	6.19 c (±0.04)	21.3 b (±0.17)
FA + SA + SM	41.13 c (±1.5)	16.9 b (±0.6)	90.4 e (±3.81)	2.37 b (±0.03)	0.27 a (±0.00)	6.3 c (±0.00)	21.4 b (±0.00)
^1^ Sig.	*******	*******	*******	******	******	*******	*******

^1^ Sig.: significance; ** and *** display the significance at 1% and 0.5% by least significant difference. Values with different roman letters (a–d) in the same row are significantly different according to the Tukey range test (*p* < 0.05).

**Table 7 molecules-25-03640-t007:** Physicochemical parameters for ciders fermented by cider yeast (C), with the following additions: SA—ammonium sulfate, SM—magnesium sulfate, FA—ammonium phosphate.

	Free Amino Nitrogen (FAN) (mg/L)	Total Polyphenol Content (mg of Catechina/100 mL)	Antioxidant Activity (mg of Trolox/ 100 mL)	Total Acidity (g of Malic Acid/L)	Volatile Acidity (g Acetic Acid/L)	Ethyl Alcohol Content (%)	Total Extract Content (g/L)
Control	48.8 a (±0.81)	14.2 a (±0.5)	70.3 a (±0.8)	1.23 a (±0.18)	0.67 (±0.34)	6.56 a (±0.19)	16.03 a (±0.76)
SA	58.2 a (±2.5)	16.3 a (±0.31)	76.8 b (±3.1)	1.93 b (±0.27)	0.33 (±0.13)	5.86 b (±0.04)	19.7 b (±1.21)
SM	50.6 a (±4.2)	14.7 a (±1.2)	74.1ab(±1.7)	1.38 ab (±0.1)	0.24 (±0.03)	5.92 b (±0.08)	19.13 b (±0.56)
FA	63.6 ab (±9.7)	15.6 a (±0.4)	77.3 b (±0.7)	1.54 ab (±0.39)	0.35 (±0.27)	5.79 b (±0.04)	18.43 b (±0.4)
SA + SM	47.8 ac (±9.22)	15.7 a (±0.93)	81.1 c (±2.9)	1.58 ab (±0.29)	0.2 (±0.08)	6.07 b (±0.07)	19.3 b (±1.14)
SA + FA	55.9 a (±2.6)	17.3 b (±1.12)	82.2 c (±1.1)	1.82 ab (±0.33)	0.19 (±0.07)	5.97 b (±0.09)	21.03 c (±0.12)
SM + FA	53.4 a (±1.9)	15.6 a (±0.6)	86.2 d (±1.9)	1.4 ab (±0.12)	0.33 (±0.13)	6.24 c (±0.05)	18.93 b (±0.11)
FA + SA + SM	48.03 ac (±4.3)	16.7 b (±1.13)	87.2 d (±0.6)	1.27 ab (±0.12)	0.23 (±0.05)	6.17 c (±0.04)	19.9 b (±0.17)
^1^ Sig.	*****	*****	*******	*****	**ns**	*******	*******

^1^ Sig.: significance; * and *** display the significance at 5% and 0.5% by least significant difference; ns: not significant. Values with different roman letters (a–d) in the same row are significantly different according to the Tukey range test (*p* < 0.05).

**Table 8 molecules-25-03640-t008:** Physicochemical parameters for ciders fermented by wild yeast (D), with the following additions: SA—ammonium sulfate, SM—magnesium sulfate, FA—ammonium phosphate.

	Free Amino Nitrogen (FAN) (mg/L)	Total Polyphenol Content (mg of Catechina/100 mL)	Antioxidant Activity (mg of Trolox/100 mL)	Total Acidity (g of Malic Acid/L)	Volatile Acidity (g Acetic Acid/L)	Ethyl Alcohol Content (%)	Total Extract Content (g/L)
Control	42.3 a (±9.9)	11.8 (±0.46)	57.2 a (±1.53)	1.49 (±0.12)	0.47 (±0.07)	6.61 a (±0.00)	16.43 a (±0.75)
SA	51.9 b (±2.8)	12.01 (±0.62)	67.3 b (±1.6)	1.73 (±0.28)	0.42 (±0.13)	5.92 b (±0.00)	18.7 b (±0.17)
SM	42.9 a (±1.8)	12.2 (±0.3)	52.8 a (±2.21)	1.68 (±0.08)	0.55 (±0.08)	5.86 b (±0.05)	19.93 b (±0.55)
FA	54.6 b (±3.1)	13.1 (±0.57)	62.8 ab (±0.6)	1.75 (±0.1)	0.41 (±0.06)	5.86 b (±0.05)	17.93 a (±1.09)
SA + SM	52.6 b (±2.5)	12.7 (±0.19)	57.3 a (±3.43)	1.67 (±0.2)	0.76 (±0.27)	5.92 b (±0.00)	18.5 b (±0.5)
SA + FA	51.7 a (±0.6)	13.13 (±0.76)	76.6 b (±1.7)	1.73 (±0.05)	0.35 (±0.09)	5.94 b (±0.12)	19.00 b (±0.00)
SM + FA	46.9 a (±2.2)	13.18 (±0.56)	67.3 b (±2.49)	1.54 (±0.13)	0.42 (±0.05)	6.24 c (±0.05)	19.66 b (±0.12)
FA + SA + SM	48.4 a (±7.0)	12.98 (±0.38)	81.5 b (±4.2)	1.79 (±0.13)	0.7 (±0.28)	6.27 c (±0.05)	18.93 b (±0.12)
^1^ Sig.	*****	**ns**	*******	**ns**	**ns**	*******	*******

^1^ Sig.: significance; * and *** display the significance at 5% and 0.5% by least significant difference; ns: not significant. Values with different roman letters (a–d) in the same row are significantly different according to the Tukey range test (*p* < 0.05).

**Table 9 molecules-25-03640-t009:** Volatile compounds in ciders fermented by distiller’s yeast (G), with the following additions: SA—ammonium sulfate, SM—magnesium sulfate, FA—ammonium phosphate.

Compound (mg/L)	Control	SA	SM	FA	SA + SM	SA + FA	SM + FA	SA + SM + FA	^1^Sig.
Ethyl acetate	80.8 a (±10.6)	50.6 b (±16.34)	39.4 b (±5.17)	55.3 bc (±6.02)	34.2 b (±1.62)	40.3 b (±0.97)	33.7 bd (±0.96)	33.5 bd (±0.6)	***
Isobutyl acetate	1.4 a (±0.03)	1.5 ab (±0.08)	1.7 c (±0.01)	1.6 c (±0.03)	1.7 c (±0.05)	1.6 bc (±0.02)	1.7 c (±0.08)	1.7 c (±0.04)	***
Ethyl lactate	n.d.	17.9 b (±1.6)	8.8 c (±0.5)	4.9 d (±0.71)	8.4 c (±0.73)	7.2 c (±1.8)	7.5 c (±0.6)	7.6 c (±0.7)	***
Isopentyl acetate	0.5 a (±0.08)	3.9 b (±0.16)	5.01 c (±0.4)	0.6 a (±0.11)	4.9 c (±0.18)	1.7 d (±0.43)	2.03 e (±0.02)	3.22 f (±0.08)	***
Ethyl hexanoate	0.27 a (±0.05)	0.45 b (±0.06)	0.5 b (±0.01)	0.54 b (±0.03)	0.01 c (±0.00)	0.16 a (±0.07)	0.7 c (±0.12)	0.00 d (±0.00)	***
Hexyl acetate	0.11 a (±0.01)	0.14 ac (±0.002)	n.d.	0.19 b (±0.02)	0.5 d (±0.002)	0.16 cd (±0.005)	0.04 b (±0.004)	0.43 e (±0.03)	***
Isobutanol	169.6 a (±17.5)	170.3 a (±34.4)	n.d.	112.9 b (±19.5)	n.d.	135.3 bc (±6.7)	190 d (±6.9)	n.d.	***
3-methyl-butanol	245.8 a (±3.2)	167.4 b (±9.3)	232.1 a (±2.4)	130.6 c (±6.7)	219.5 ac (±15.6)	128.7 c (±2.9)	218.4 a (±3.8)	210.16 d (±3.8)	***
2-methyl-butanol	171.3 a (±2.3)	124.6 b (±2.3)	163.7 a (±7.16)	101.3 c (±4.19)	159.3 a (±15.3)	100.7 c (±3.03)	150.17 d (±5.19)	139.7 d (±27.1)	***
Diethyl acetal	2.3 a (±0.07)	n.d.	2.7 a (±0.03)	2.64 a (±0.5)	2.5 a (±0.05)	n.d.	3.33 b (±0.7)	3.15 ab (±0.03)	***

^1^ Sig.: significance; *** displays the significance at 0.5% by least significant difference. Values with different roman letters (a–f) in the same row are significantly different according to the Tukey range test (*p* < 0.05).

**Table 10 molecules-25-03640-t010:** Volatile compounds in ciders fermented by wine yeast (W), with the following additions: SA—ammonium sulfate, SM—magnesium sulfate, FA—ammonium phosphate.

Compound [mg/L]	Control	SA	SM	FA	SA + SM	SA + FA	SM + FA	SA + SM + FA	^1^Sig.
Ethyl acetate	55.8 a (±4.84)	33.16 b (±0.87)	22.5 c (±0.68)	43.2 d (±1.17)	27.3 bc (±0.74)	33.7 a (±0.5)	54.4 a (±2.71)	49.25 ad (±3.6)	***
Isobutyl acetate	1.45 a (±0.02)	1.6 b (±0.04)	1.5 ab (±0.04)	1.6 b (±0.01)	1.73 c (±0.02)	1.61 b (±0.02)	1.73 c (±0.05)	1.67 c (±0.02)	***
Ethyl lactate	n.d.	5.6 a (±0.84)	n.d.	1.05 b (±1.83)	4.27 a (±0.68)	5.42 a (±0.35)	2.08 b (±1.82)	4.47 a (±0.8)	***
Isopentyl acetate	0.16 a (±0.01)	1.2 b (±0.33)	0.3 a (±0.014)	0.24 a (±0.014)	0.7 ab (±0.35)	0.89 b (±0.07)	0.28 a (±0.09)	0.61 a (±0.25)	***
Ethyl hexanoate	0.3 a (±0.05)	0.5 b (±0.09)	0.5 b (±0.03)	0.7 c (±0.01)	n.d.	0.22 a (±0.02)	0.32 a (±0.09)	n.d.	***
Hexyl acetate	0.13 a (±0.01)	0.19 b (±0.03)	n.d.	0.21 b (±0.01)	0.56 c (±0.03)	0.2 b (±0.01)	0.15 ab (±0.01)	0.7 c (±0.04)	***
Isobutanol	61.9 a (±2.3)	49.3 b (±2.8)	n.d.	45.05 bc (±2.3)	n.d.	43.5 c (±1.14)	71.2 d (±3.02)	n.d.	***
3-methyl-butanol	154.4 a (±1.3)	97.6 b (±1.7)	138.8 c (±3.4)	82.7 d (±1.3)	137.8 c (±0.95)	80.2 d (±0.7)	135.1 c (±3.5)	123.5 e (±0.6)	***
2-methyl-butanol	109.5 a (±1.12)	73.2 b (±1.72)	100.5 c (±2.8)	65.8 d (±0.7)	104.5 c (±1.6)	62.8 d (±0.4)	103.4 c (±2.4)	96.6 d (±1.7)	***
Diethyl acetal	2.6 a (±0.1)	1.18 a (±2.05)	2.6 a (±0.6)	0.15 b (±0.1)	2.9 a (±0.5)	n.d.	3.34 a (±0.12)	3.35 a (±0.15)	***

^1^ Sig.: significance; *** displays the significance at 0.5% by least significant difference. Values with different roman letters (a–e) in the same row are significantly different according to the Tukey range test (*p* < 0.05).

**Table 11 molecules-25-03640-t011:** Volatile compounds in ciders fermented by wild (D) yeast, with the following additions: SA—ammonium sulfate, SM—magnesium sulfate, FA—ammonium phosphate.

Compound [mg/L]	Control	SA	SM	FA	SA + SM	SA + FA	SM + FA	SA + SM + FA	^1^Sig.
**Ethyl acetate**	41.8 a (±3.8)	60.4 a (±6,4)	38.1 a (±0.75)	55.8 a (±2.4)	86.13 bc (±9.6)	71.2 bc (±10.7)	89.12 c (±12.9)	67.6 b (±2.3)	***
**Isobutyl acetate**	1.5 a (±0.03)	1.5 a (±0.05)	1.7 b a (±0.04)	1.5 a (±0.03)	1.74 b (±0.02)	1.7 b (±0.02)	1.74 b (±0.01)	1.66 b (±0.06)	***
**Ethyl lactate**	n.d.	2.6 (±1.2)	n.d.	2.8 (±0.05)	n.d.	3.5 (±3.9)	4.6 (±3.9)	1.06 (±0.02)	ns
**Isopentyl acetate**	0.14 (±0.01)	0.3 (±0.02)	0.3 (±0.04)	0.3 (±0.02)	0.4 (±0.07)	0.3 (±0.3)	0.2 (±0.02)	0.31 (±0.04)	ns
**Ethyl hexanoate**	0.17 a (±0.04)	0.3 b (±0.03)	0.3 b (±0.03)	0.6 c (±0.01)	n.d.	0.17 a (±0.04)	0.26 d (±0.04)	0.06 ba (±0.04)	***
**Hexyl acetate**	0.1 a (±0.01)	0.16 b (±0.01)	n.d.	0.17 b (±0.01)	0.5 c (±0.02)	0.15 b (±0.01)	0.12 a (±0.01)	0.5 c (±0.01)	***
**Isobutanol**	75.1 ac (±3.3)	73.7 a (±9.80	88.5 a (±6.8)	53.2 bc (±4.4)	71.8 ac (±0.7)	52.5 bc (±2.9)	74.1 ac (±5.2)	56.9 c (±2.0)	***
**3-methyl-butanol**	139.2 a (±6.8)	98.8 b (±3.2)	141.4 a (±0.4)	88.13 c (±1.14)	117.8 d (±1.8)	85.9 c (±2.03)	118.4 d (±0.9)	107.7 e (±20.3)	***
**2-methyl-butanol**	91.3 a (±2.1)	69.5 b (±0.95)	96.7c (±0.78)	60.8 d (±1.27)	82.7 e (±0.14)	61.5 d (±0.99)	85.4 e (±1.29)	77.12 f (±2.62)	***
**Diethyl acetal**	2.7 a (±0.1)	0.76 b (±0.67)	3.03 a (±0.12)	0.38 b (±0.29)	3.07 a (±0.14)	0.69 b (±0.09)	3.78 ac (±0.04)	3.44 a (±0.17)	***

^1^ Sig.: significance; *** displays the significance at 0.5% by least significant difference; ns: not significant. Values with different roman letters (a–f) in the same row are significantly different according to the Tukey range test (*p* < 0.05).

**Table 12 molecules-25-03640-t012:** Volatile compounds in ciders fermented by cider yeast (C), with the following additions: SA—ammonium sulfate, SM—magnesium sulfate, FA—ammonium phosphate.

Compound [mg/L]	Control	SA	SM	FA	SA + SM	SA + FA	SM + FA	SA + SM + FA	^1^Sig.
Ethyl acetate	39.6 a (±17.8)	31.8 a (±0.98)	23.4 a (±2.5)	39.4 a (±0.15)	28.4 a (±1.1)	40.2 a (±6.5)	37.5 a (±2.7)	43.3 b (±0.19)	*
Isobutyl acetate	1.28 a (±0.04)	1.4 b (±0.01)	1.6 c (±0.03)	1.5 c (±0.02)	1.7 de (±0.03)	1.5 c (±0.01)	1.58 ce (±0.05)	1.65 e (±0.06)	***
Ethyl lactate	n.d.	10.9 (±9.4)	1.2 (±2.04)	2.9 (±0.39)	3.4 (±0.001)	4.18 (±0.3)	4.16 (±0.4)	0.38 (±0.65)	ns
Isopentyl acetate	0.37 (±0.25)	0.6 (±0.53)	0.5 (±0.18)	0.3 (±0.12)	0.54 (±0.07)	0.56 (±0.31)	0.58 (±0.28)	0.21 (±0.04)	ns
Ethyl hexanoate	0.55 a (±0.04)	0.4 a (±0.35)	0.5 a (±0.21)	0.8 b (±0.01)	0.31 a (±0.05)	0.29 a (±0.21)	0.85 b (±0.04)	0.03 a (±0.13)	*
Hexyl acetate	0.16 (±0.005)	0.21 (±0.01)	n.d.	0.21 (±0.002)	0.19 (±0.01)	0.19 (±0.01)	0.18 (±0.00)	4.8 (±7.1)	ns
Isobutanol	58.7 (±3.6)	64.10 (±3.8)	n.d.	44.9 (±2.7)	n.d.	63.18 (±1.6)	77.4 (±12.3)	37.7 (±53.3)	ns
3-methyl-butanol	154.2 ae (±2.2)	114.5 b (±1.03)	158.7 e (±2.2)	89.7 c (±1.8)	149.8 ae (±11.8)	107.9 b (±2.6)	137.4 af (±7.04)	135.8 df (±9.02)	***
2-methyl-butanol	97.04 a (±2.2)	76.9 b (±1.56)	109.6 a (±1.6)	65.5 b (±0.66)	105.3 a (±8.2)	77.7 b (±1.45)	99.7 a (±5.8)	100.8 a (±9.3)	***
Diethyl acetal	1.92 a (±0.05)	1.2 a (±0.5)	2.5 a (±0.04)	1.7 a (±0.5)	n.d.	3.8 b (±0.6)	3.2 b (±0.35)	3.7 b (±0.51)	***

^1^ Sig.: significance; * and *** display the significance at 5% and 0.5% by least significant difference; ns: not significant. Values with different roman letters (a–f) in the same row are significantly different according to the Tukey range test (*p* < 0.05).

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
