# Peer review of "The Quality of Ciders Depends on the Must Supplementation with Mineral Salts"

_molecules, 2020, doi:10.3390/molecules25163640_

Round 1
Reviewer 1 Report
Authors refer to the impact of mineral salts [(NH4)2SO4, MgSO4, (NH4)3PO4] supplementation in apple must checking their influence on oenological parameters, antioxidant activity, total polyphenol content and evaluating the profile of volatile cider compounds fermented with various yeast strains.
In my opinion, the study is well organized and the results and discussion are properly described.
I have just few questions
- Salts concentrations reported in the abstract are different from the ones reported in section 2.1.1
- Why authors used a concentration rate of (NH4)2SO4, MgSO4, (NH4)3PO4 between 0.5-0.167 g/L. Usually the amount used for these salts, for improving fermentation process, are higher.
- Authors refers to glycerol production during the fermentation process but it is not reported in tables. Usually, salts addition is followed by ethanol production increasing and glycerol decreasing since salts could promote the NADH re-oxidation by supporting different cellular metabolisms.
- I guess authors mean MgSO4 7H2O
- Finally, did the authors consider the initial glucose concentration of the must?
Reviewer 2 Report
I have carefully read manuscript molecules-893222 entitled „The quality of ciders depends on the must supplementation with mineral salts“. The scope of the paper is of interest and I have found a general good quality of the research. From experimentation to data evaluation, everything is well organized and clearly described and the all analysis appears to be carefully performed. I think that this work may be the subject of some future research related to the preparation of various alcoholic beverages. In my opinion, this work could be accepted to be published in Molecules after minor revision. I have only few remarks:
- Did the authors try to do any statistical analysis of the obtained experimental data?
- In your opinion, what are the best conditions (salt concentrations and which yeast) for obtaining a quality cider?
- Did the author take into consideration how the addition of salt will affect the taste of the drink that is the final product? Maybe the bitterness is changing?
